# Effects of Virtual Reality-Based Graded Exposure Therapy on PTSD Symptoms: A Systematic Review and Meta-Analysis

**DOI:** 10.3390/ijerph192315911

**Published:** 2022-11-29

**Authors:** Seoyoon Heo, Jin-Hyuck Park

**Affiliations:** 1Department of Occupational Therapy, College of Medical and Health Science, Kyungbok University, Namyangju-si 42517, Republic of Korea; 2Department of Occupational Therapy, College of Medical Science, Soonchunhyang University, Asan-si 31538, Republic of Korea

**Keywords:** exposure therapy, posttraumatic stress disorder, virtual reality, meta-analysis

## Abstract

Previous studies reported that virtual reality (VR)-based exposure therapy (VRET) was a clinically beneficial intervention for specific phobias. However, among VRET, VR-based graded exposure therapy (VR-GET) is little known about its efficacy on posttraumatic stress disorder (PTSD) symptoms. Therefore, this meta-analysis investigated the effects of VR-GET for PTSD symptoms. A literature search yielded seven randomized controlled trials. The differences between conditions regarding the primary outcome of PTSD symptoms in the effect size of the individual study were calculated using Hedges’ g. The findings showed VR-GET showed a significantly larger effect size for PTSD symptoms (*g* = 1.100, *p* = 0.001), compared to controls. However, no significant difference between conventional VRET and controls was found for PTSD symptoms (*g* = −0.279, *p* = 0.970). These findings indicated the superiority of VR-GET for PTSD symptoms compared to controls, supporting the importance of immersive PTSD treatments. Nevertheless, the results need to be interpreted with caution due to the substantial number of military service personnel studies. Future trials, considering individually tailored scenarios in virtual environments to cover a wider range of trauma types, are required to investigate its evidence on treating PTSD.

## 1. Introduction

Posttraumatic stress disorder (PTSD) is a prevalent mental health issue that results from experiencing or being exposed to a traumatic event such as death, combat, or severe injury [1,2]. Thus, a variety of traumatic events such as domestic violence, sexual assault, or motor-vehicle accidents could lead to PTSD [1,2]. Among these events, warfare is one of the most challenging events as it demands a high level of physical, cognitive, and psychological efforts [3]. Accordingly, war veterans exposed to military combat are at higher risk of PTSD and have experienced PTSD [3,4].

People diagnosed with PTSD have a tendency to show persistent symptoms and an unremitting chronic course [1,5], which is closely associated with high social and healthcare costs. Specifically, in a sample of United States (US) veterans, the social costs were estimated at $923 million for two years [2,6,7]. Given these high social costs, effective and timely intervention needs to be established.

Various types of psychological interventions have been implemented to ameliorate PTSD and its related symptoms [1,2]. A review study indicated that trauma-focused cognitive-behavioral therapy (CBT), cognitive processing therapy (CPT), eye movement desensitization and reprocessing (EMDR), and exposure therapy were effective for PTSD [1,2]. Among these interventions, exposure therapy is regarded as the first-line treatment for PTSD with abundant evidence of its clinical efficacy [4,7].

However, despite the effectiveness of psychological interventions including exposure therapy, a major disadvantage lies in the difficulty in fully immersing subjects in traumatic scenes [4]. This disadvantage could cause high dropout rates, negatively affecting interventions effects [8]. Therefore, PTSD remains a difficult disorder to treat, and needs for alternative intervention approaches are needed.

To address these issues, exposure therapy using virtual reality (VR) has been highlighted as having potential efficacy in the treatment of PTSD because it can compensate for the shortcomings of existing psychological interventions [9]. VR can offer multi-sensory stimuli tailored to a patient’s individual trauma in a highly interactive, ecologically valid, and emotionally engaging virtual environment, which recapitulates a patient’s traumatic event with computer-generated visual, auditory, olfactory, and haptic experiences for therapeutic purposes [10]. This feature could contribute to the implementation of VR as a novel method for exposure therapy. Indeed, a prior meta-analysis study reported that VR-based exposure therapy (VRET) yielded a lower dropout rate than conventional exposure therapy [11].

VRET presents immersive sensory cues in digital surroundings, and subjects can interact with stimuli using VR devices such as a head-mounted visual display. VR depicts fear stimuli realistically to address trauma while allowing the clinician to freely manipulate the stimuli for therapeutic purposes. Therefore, VR is less dependent on the subject’s imagination and the clinician’s effort to simulate trauma [12]. With these advantages, VRET used for patients with PTSD has been identified as effective. In a pilot study, subjects who were exposed to a VR war scenario showed considerable statistical reduction of depression and anxiety, suggesting that VRET could be promising intervention for patients with combat-related PTSD [13]. Another study reported that VRET used in the treatment of World Trade Center attack-related PTSD was found to effective [14]. Similarly, randomized controlled trials found that VRET considerably reduced PTSD symptoms compared to other treatments [13,15,16,17,18,19], implying that the use of VRET could be promising for patients with PTSD.

However, unlike the findings of individual studies reporting the positive effects of VRET, comprehensive evidence on the efficacy of VRET for PTSD is still inconclusive [2]. Previous studies evaluating the effectiveness of VRET for PTSD symptoms involved varying methodologies and samples, resulting in mixed results. Specifically, randomized controlled studies showed no significant difference in the efficacy between VRET and conventional interventions [19,20].

This inconsistency could be attributed to the fact that some patients with PTSD could not be fully immersed as VR-based exposure therapy progressed [19,20], and this plays a crucial role in its efficacy. Unfortunately, in conventional VRET, the intensity and frequency of trauma-related stimuli are graded as the training session progresses, not according to the subject’s response [13,19,20]. However, this method has less chance of fully immersing patients with PTSD as trauma-related stimuli are not presented in consideration of the patient’s response to them.

To address this issue, in previous studies, to fully immerse subjects with PTSD to virtual environments in a graded fashion, different levels of physiological responses or subjective distress were used to control the frequency and intensity of trauma-related stimuli during VRET sessions rather than automatically grading them as the treatment sessions progressed [15,16,17]. This VR-based graded exposure therapy (VR-GET) immerses patients in a VR environment while monitoring their responses, allowing therapists to learn what stimuli could trigger their response and assess the efficacy of the levels of VR exposure [21], which is the advantage compared to conventional VRET. Thus, a meta-analysis of randomized controlled trials of VR-GET should be conducted by analyzing its effects compared to conventional VRET.

The objective of the current meta-analysis was to investigate the differences in the effects between VR-GET and conventional VRET in patients with PTSD compared to control conditions. This study hypothesized that VR-GET would have a considerable positive effect on PTSD symptoms compared to control conditions.

## 2. Materials and Methods

This systematic review and meta-analysis followed the Preferred Reporting Items for Systematic Reviews and Meta-analysis statement (PRISMA) [22].

### 2.1. Search Strategy and Study Selection

A literature search was completed in May 2022. This search focused on trials published from January 2010 to April 2022 Google Scholar, PubMed, Embase, Web of Science, Cochrane Central Register of Controlled Trials, PsyInfo, and MEDLINE electronic databases. The search terms were “PTSD” or “Trauma” and “VR” or “Virtual reality” and “Exposure” or “Exposure therapy” or “Exposure treatment” or “exposure intervention”. Only English-language papers were included.

The initial eligibility screening was independently conducted by two reviewers based on the titles and abstracts of the searched trials. Disagreements between the reviewers were settled by a discussion with an additional reviewer.

### 2.2. Eligibility Criteria

To be eligible for inclusion, studies needed to comply with the following criteria:Study design: randomized controlled trials (RCTs)Population: patients with PTSD according to the Diagnostic and Statistical Manual of Mental Disorder-fourth edition (DSM-4) or DSM-5 criteria for PTSDIntervention: (a) VRET was used as a treatment to reduce PTSD symptoms, (b) VRET minimally consisted of the use of HMD or a large projector screen, and (c) VRET was implemented with minimal guidance such as training provided by a clinicianControl: the controls received at least one “non-VR-based” exposure therapy, including cognitive behavior therapy, prolonged exposure, present-centered group therapy or any other conventional treatments or were treated as usual or minimal attention such as brief telephone contactsOutcomes: (a) the primary outcomes were pre- and post-test measures of PTSD symptom severity and (b) the secondary outcomes included the symptoms of depression

### 2.3. Quality Assessment

The Cochrane risk of bias assessment tool was used to assess the methodological quality of the RCTs and the risk of bias. If the studies were identified as having a high or unclear risk of bias for the assessor’s blinding or incomplete outcome data sections, they were considered to have a high risk of bias [23]. Two reviewers independently evaluated the risk of bias, and both reviewers established consensus scores after the discussion of any disparate assessment.

### 2.4. Data Extraction

The following data were extracted in duplicate using data extraction forms: (1) characteristics of subjects (size, mean age and diagnostic information); (2) intervention features (duration, intervention components, and type of controls); and (3) effects on PTSD symptoms (outcome measures and changes in PTSD symptoms). All outcome measures were recorded as means, standard deviations, *p*-values, and t-values of F-values for each group at pre- and post-test or follow-up test. If a trial did not present enough data, its corresponding authors were contacted to retrieve the data. If the authors did not respond with sufficient data, the studies were excluded. Disagreements between both reviewers were addressed via a discussion.

### 2.5. Statistical Analysis

All analyses were conducted using Comprehensive Meta-Analysis 2.0 (Biostat, Englewood, NJ, USA). Statistical heterogeneity, effect size, and publication bias of the selected studies were analyzed. Hedges’ g was calculated to derive the standardized mean differences with 95% confidence intervals (CIs). Pooled Hedges’ g estimates of <0.30, ≥0.30 and <0.60, and ≤0.60 represented small, moderate, and large effect sizes, respectively [24]. I^2^ statistics were used to test heterogeneity among the studies. An I^2^ > 50% with a *p*-value of ≤ 0.05 indicated notable heterogeneity, and a random effect model was used. Otherwise, a fixed effect model was used [25]. Publication bias was visually analyzed using funnel plots and Egger’s regression intercept test [26].

## 3. Results

### 3.1. Study Selection

A total of 496 studies were identified in the initial literature review. Among them, 124 duplicate articles were removed. The title and abstract of the remaining 372 articles were reviewed for preliminary screening. Of these, eight articles that met the inclusion criteria were finally selected (Figure 1).

### 3.2. Characteristics of the Included Studies

A total of 308 subjects with ages ranging from 28.0 to 58.0 years were included. The subjects in these studies were military soldiers on active duty or veterans with combat-related PTSD that met DSM-4 or DSM-5 criteria. VRET mostly used simulation with specific or unspecific wartime environment conditions, and the number of VRET sessions ranged from 6 to 14 sessions that lasted 70–90 min. Among VRETs, cases when the VR environment was controlled by continuously monitoring the subject’s responses, such as physiologic responses or distress levels, were classified as VR-GET. Otherwise, those that included an automatic adjustment of the VR environment or trauma-related stimuli as treatment sessions progressed and did not consider the subject’s responses were classified as conventional VRET. The controls underwent conventional exposure therapy, present-centered therapy, and non-trauma-focused treatment or received a treatment as usual and minimal attention.

PTSD symptoms were measured as the primary outcome in all studies. The following tools were used: the Clinician Administered PTSD Scale (CAPS), the PTSD Checklist (PCL), the Posttraumatic Avoidance Behavior Questionnaire (PABQ), the Impact of Events Scale Revised (IES-R), and the Trauma-related Guilt Inventory (TRGI). Detailed information is shown in Table 1.

### 3.3. Risk of Bias Assessment

Results from the Cochrane Risk of Bias assessments indicated that there was performance and detection bias in the included studies, with only two of seven studies using a blind method in which the assessors were not aware of the participants’ treatment status (Figure 2).

### 3.4. Effect Size of VR-Based Exposure Therapy on PTSD Symptoms

Effect sizes were significantly greater when comparing VR-GET to controls than when comparing conventional VRET to controls (Figure 3). When compared to controls, VR-GET showed a large positive effect size (*g* = 1.10, 95%, *p* = 0.001), suggesting that VR-GET was more beneficial in reducing PTSD symptoms, compared to controls. In contrast, the meta-analysis revealed no significant difference in PTSD symptoms between conventional VRET and controls (*g* = −0.279, *p* = 0.970). There was no considerable heterogeneity in the study data (conventional VRET: *p* = 0.306, I^2^ = 17.04%; VR-GET: *p* = 0.083, I^2^ = 59.76%), so a fixed-effects model was used to evaluate effect sizes. The funnel plot did not show significant asymmetry (Egger’s intercept = 3.119, *p* = 0.051), suggesting no considerable publication bias (Figure 4).

### 3.5. Meta-Regression Analysis of VR-Based Exposure Therapy Sessions

The meta-analysis showed a trend in the dose–response relationship, but there was no significant dose effect (*p* = 0.159) (Figure 5), which indicated that more therapy sessions could not yield larger effect sizes.

## 4. Discussion

Considering the needs for stronger evidence on the value of using VR for PTSD intervention, this study was conducted to systematically review controlled trials and to analyze the effects of VR-GET. The findings revealed that VR-GET could be a promising intervention for improving PTSD symptoms. The effect size for PTSD symptoms reduction in favors of VR-GET was large (*g* = 1.10), without significant heterogeneity and no indication of publication bias. These findings are consistent with recent meta-analyses reporting the positive effects of VRET [2,4], but depict better outcomes [2,4,12] as the current meta-analysis concluded the efficacy of VRET by limiting it to VR-GET, which was the main original feature discriminating the findings of this study compared to prior meta-analyses. In contrast, our results indicated that no significant difference between conventional VRET and controls in improving PTSD symptoms (*g* = −0.279, *p* > 0.05). These results suggest the importance of VRET immersion in its therapeutic effect.

With the development of VR technology, the use of VR has been highlighted in psychological treatments. Three recent meta-analyses on the efficacy of VR-based exposure therapy were implemented within three years, which might suggest the increased interest in using VR technology to improve PTSD symptoms [2,4,12,28].

The current study revealed that overall VRET had no comparable effect size on PTSD symptoms. Similarly, in previous reviews on other technological interventions for mental health, no considerable differences between the effectiveness of technological interventions and conventional interventions were found [2,4,29], which is consistent with our findings. Thus, there were no head-to-head advantages of VRET versus control treatments.

However, VR-GET was found to be more beneficial in ameliorating PTSD symptoms compared to controls. In contrast, conventional VRET did not show considerable effect on PTSD symptoms. This disparity could be attributed to VR-GET immersion. In VR-GET, the subjects could fully engage in a trauma-related environment as clinicians could provide trauma-related stimuli in a gradual manner by continuously monitoring their responses. Indeed, in a prior study, the subject’s distress level measured by the Subjective Units of Distress Scale rating every five minutes during training sessions [17]. In a previous study, if the subjects did not seem to be fully immersed while describing trauma-related events, the therapists increased the intensity of the trauma-related stimuli, allowing them to become fully engaged. This factor could have maximized the effects of VR-GET. In contrast to VR-GET, in conventional VRET, the degree to which subjects experienced virtual environments to aid in fully emotional engagement was not measured, and trauma-related stimuli were not adjusted accordingly [4,13,19,20,27]. A previous study implied that the intensity and frequency of trauma-related stimuli needed to increase as the subjects adapted and habituated to their trauma-related events, supporting our findings [17], and suggesting the importance of graded exposures to trauma-related stimuli.

The meta-regression indicated no significant dose–response relationship. Thus, it could not be said that more treatment sessions would lead to better outcomes. In line with this finding, a previous review study recommended a range from 8 to 12 sessions (90 min a session), supporting the finding of this study [30]. Nevertheless, in this meta-analysis, the range of training durations (8 to 14 sessions) was not wide, so dose–response interpretations should be done carefully.

Although, no advantage of overall VRET over controls was found, given that exposure therapy is considered as first-line treatment for PTSD [31], VR could be a potentially efficacious method for exposure therapy for PTSD [2]. In general, VRET could constitute an ecologically valid and safe environment for inducing emotional reactions by presenting a variety of pre-programmed stimuli and environments. This could be an interesting option to increase a subject’s immersion. Therefore, from the point of view that VRET could be an alternative option, it is important to review whether VRET therapy is cost-effective enough. Fortunately, since the hardware and software costs of VR-based system decrease with the growth of the VR commercial market, VRET has been considered to be more cost-effective than conventional treatments requiring tremendous clinician’ efforts [32,33]. Thus, the non-inferiority of overall VRET and the superiority of VR-GET compared to controls confirmed by the current meta-analysis have the clinical implication in terms of broadening the options for PTSD treatments especially considering its low drop-out rate [11]. In particular, considering that VR-GET could be a better option than conventional VRET, it is necessary to provide graded VR simulation exposure by monitoring the subject’s reaction more sensitively.

There were a few limitations to this meta-analysis. Firstly, although PTSD could be induced by a variety of traumatic events, all of the subjects in the included studies were active-duty soldiers and veterans whose PTSD was caused by combat-related trauma. Although the problems are not the sources of heterogeneous of subjects, it is unclear whether the findings of this study are applicable to all individuals with PTSD due to a source of bias. Therefore, the efficacy of VRET should be limited to combat-related PTSD. Secondly, some studies in the included studies were identified as having a high risk of bias as they were not blined, which suggests that caution should be used in interpreting the current findings. Thirdly, due to the limited type of PTSD included in this study, it was unable to perform a subgroup analysis. Military personnel have been observed to be less responsive to PTSD treatment than other individuals with PTSD [32], which might have a certain impact on the results. Finally, due to the small number of the included studies, it was impossible to determine which subject’s response was most effective in providing graded exposures through continuous monitoring. Therefore, future studies are needed with a variety of samples to determine whether VR-GET can reduce PTSD symptoms by providing virtual individually tailored exposure experiences.

## 5. Conclusions

The current meta-analysis shed new light on the importance of immersive PTSD treatments using graded exposures. The findings suggest that VR could be a better option as it may present immersive trauma-related stimuli by monitoring the subject’s responses, such as physiological or subjective stress levels. In the future, more research on various types of PTSD is required to determine whether VRET could be considered a valuable tool for PTSD treatments.

## Figures and Tables

**Figure 1 ijerph-19-15911-f001:**
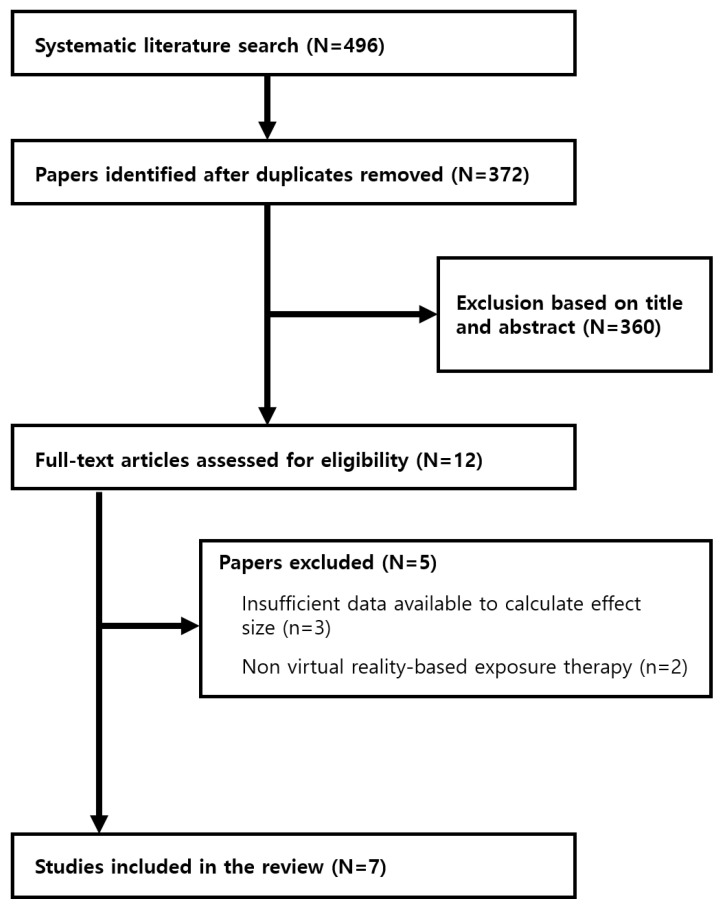
Flow chart of the study selection process.

**Figure 2 ijerph-19-15911-f002:**
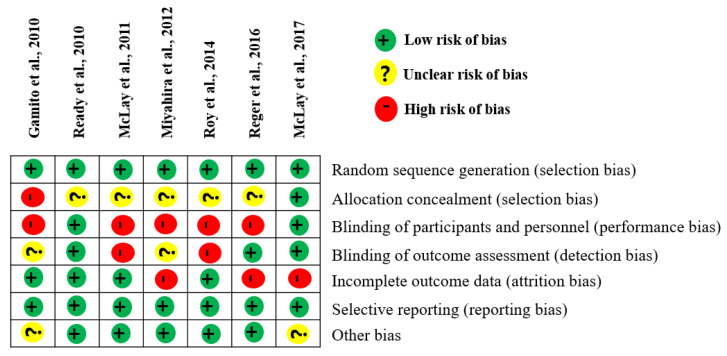
Summary of the risk of bias of the included studies in this meta-analysis. Green indicates a low risk of bias, yellow indicates unclear bias, and red indicates a high bias risk. Gamito et al. (2013) [13]; McLay et al. (2011) [15]; Miyahira et al. (2012) [16]; Ready et al. (2010) [17]; McLay et al. (2017) [19]; Reger et al. (2016) [20]; Roy et al. (2014) [27].

**Figure 3 ijerph-19-15911-f003:**
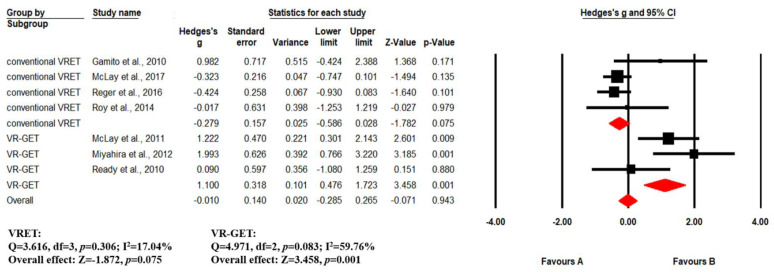
Forest plot demonstrating the efficacy of VR-based exposure therapy on PTSD symptoms. Gamito et al. (2013) [13]; McLay et al. (2011) [15]; Miyahira et al. (2012) [16]; Ready et al. (2010) [17]; McLay et al. (2017) [19]; Reger et al. (2016) [20]; Roy et al. (2014) [27].

**Figure 4 ijerph-19-15911-f004:**
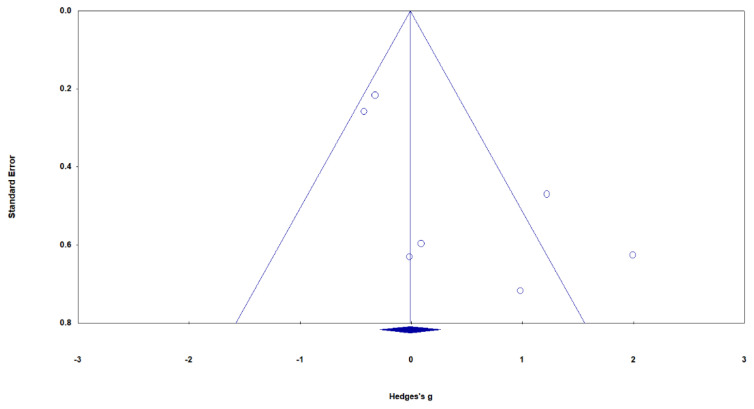
Funnel plot demonstrating the bias of VR-based exposure therapy on PTSD symptoms.

**Figure 5 ijerph-19-15911-f005:**
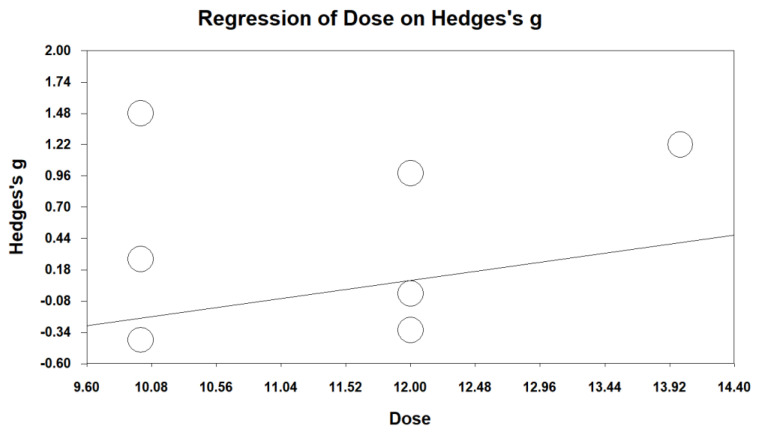
Meta-regression analysis assessing the dose effect of VR-based exposure therapy on PTSD symptoms.

**Table 1 ijerph-19-15911-t001:** Characteristics of the studies included in the systematic review and meta-analysis.

Authors	Sample Type	N	Age	VR Exposure	Duration	Control Group	Outcome Measure
VR-GET							
Ready et al., 2010 [17]	Vietnam veterans	E: 6 C: 5	E: 57 C: 58	Vietnam wartime environments with subjective distress level monitoring	10 sessions (90-min)	Present-centered therapy	CAPS
McLay et al., 2011 [15]	On active-duty soldiers	E: 10 C: 10	E: 28 C: 28.8	Iraq and Afghanistan wartime environments with physiologic monitoring	14 sessions	Treatment as usual	CAPS
Miyahira et al., 2012 [16]	On active-duty soldiers or veterans with Iraq or Afghanistan	E: 10 C: 10	E: n.p C: n.p	Unspecified wartime environments with subjective distress level monitoring	10 sessions	Minimal attention (brief telephone contacts)	CAPS, PDS, TRGI
VRET							
Gamito et al., 2010 [13]	Portugues colonial war veterans	E: 4 C: 3	E: n.p C: n.p	Unspecified wartime environments without any monitoring	12 sessions	Imaginal prolonged exposure	CAPS, BDI, IES-R, SOPI
Roy et al., 2014 [27]	On active-duty soldiers or veterans with Iraq or Afghanistan	E: 9 C: 9	E: 34.5 C: 34.1	Iraq and Afghanistan wartime environments without any monitoring	12 sessions (90-min)	Imaginal prolonged exposure	CAPS, PCL
Reger et al., 2016 [20]	On active-duty soldiers or veterans with Iraq or Afghanistan	E: 54 C: 54	E: 29.5 C: 30.9	Iraq and Afghanistan wartime environments without any monitoring	10 sessions (90~120-min)	Imaginal prolonged exposure	CAPS, PCL
McLay et al., 2017 [19]	On active-duty soldiers or veterans with Iraq or Afghanistan	E: 43 C: 38	E: 33 C: 32	Iraq and Afghanistan wartime environments without any monitoring	8~12 sessions (90-min)	Exposure therapy with computer images	CAPS

VR-GET: virtual reality-based graded exposure therapy; VRET: virtual reality-based exposure therapy; CAPS: clinician-administered posttraumatic stress disorder scale; BDI: beck depression inventory; IES-R: impact of events scale revised; SOPI: sense of presence, immersion, and cybersickness; PCL: posttraumatic stress disorder checklist score; PABQ: posttraumatic avoidance behavior questionnaire; TRGI: trauma-related guilt inventory.

## Data Availability

Not applicable.

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
