# Peer review of "Effects of Virtual Reality-Based Graded Exposure Therapy on PTSD Symptoms: A Systematic Review and Meta-Analysis"

_ijerph, 2022, doi:10.3390/ijerph192315911_

Round 1

Reviewer 1 Report (Previous Reviewer 2)

The new version of the manuscript addresses most of my previous comments, thus I recommend this paper for publication.

Author Response

Thank you for your comments. We have tried to revise our manuscript in accordance with your comments as much as possible.

Reviewer 2 Report (New Reviewer)

The manuscript presents a SysRev with Meta-analysis for virtual reality-based graded exposure therapy on PTSD symptoms. This is a significant contribution to the field at a time when it becomes essential to identify strategies that promote better clinical outcomes for PTSD patients, namely military personnel.

I fail to understand how a manuscript can be submitted with a significant part of the text in red. Not only it seems odd, as writing in red is truly unpolite (unless it is related to track changes, but even those can be changed in word settings to another color). The reviewer urges the authors to refrain from repeating this in future submissions. 

The major drawback of the paper at this stage is the quality of the English. Here are some examples to illustrate this since, as a reviewer, I will not conduct an extensive sentence-by-sentence review of the manuscript:

28 -  In particular, war veterans who (or "who were") exposed to military combat are at higher risk of PTSD and have experienced PTSD (Can't understand what the authors wanted to convey with this last portion of the sentence)  [3, 4].

30 - People diagnosed with PTSD have a tendency that they to show persistent symptoms and an unremitting chronic course [1, 5], it which is closely associated with high social and healthcare costs.

35 - Various types of psychological interventions have been implemented to ameliorating ameliorate PTSD and related symptoms

38 - ...sitization and reprocessing (EMDR), and exposure therapy are efficacy effective for PTSD [1, 2].

42 - ...sure therapy, a major disadvantage lies in it is the difficulty for subjects to fully immerse in traumatic scenes has been posed.

50 - ..., which augment a patient’s trauma event with computer-generated visual, auditory, olfactory, and haptic experiences [10]. - We really hope it doesn't augment the trauma of the patients, which would render VRET useless in PTSD.

56 - VR depicts fear stimuli in realistic ways realistically to bring certain address trauma while allowing the clinician to freely manipulate the stimuli for therapeutic purposes.  and these stimuli can be manipulated freely by a clinician.

Introduction

The authors address some of the literature, but it is highly recommended they also address non-RCT studies which make for most of the evidence in this field. These may not be included in the SysRev but should be addressed as evidence in the introduction. Authors such as Rizzo. Wierderhold and Gamito had some initial promising work in this context, namely related to War PTSD.

Authors should also consider the importance of PTSD following other traumatic events, such as domestic violence, sexual assault of motor-vehicle accidents.

Authors should explain in a more detailed fashion the differences between VRET and VR-GET, especially in PTSD. From my experience with these patients, only VRET with gradually increasing presentation of stimuli should be used. 

Also, why is the focus on VR-GET if most of the papers used in the review classify their interventions as VRET? 

Methods

Would recommend that the authors include the PICO tool. It makes it easier for the reader and specialists when analyzing the manuscript. 

Results

138 - 124 duplicate 138 articles were removed - Do not start sentences with numbers.

139 - And then we read the title and abstract of the remaining 372 articles for preliminary screening. - Always try to write this in a way that uses mostly third-person. Avoid using first-person.

Please ensure that the classification of the intervention is correct. For Gamito et al, for instance, the reviewer is not sure if the study can be classified as a pure RCT. Also, since this review used the term VR-GET, if the authors check the methods section of Gamito et al, you can find "Both the VRET and EI conditions maintained the regular psychotherapy rationale based on cognitive desensitization with 12 graded exposure sessions of VR for the VRET condition and 12 sessions of imagination exposure for the EI condition." In the Ready et al study there is no mention of VR-GET.

This is a serious concern as it impacts in the interpretation of the results and the conclusions included in the discussion.

Discussion

As mentioned above, the conclusions are inaccurate since the classification of the studies and the resulting outcomes are, in the view of the review, incorrect. The reviewer strongly suggests to the authors a revision of the classification of the studies in VRET and VR-GET. The study from Gamito comes to mind, but the misclassification can include others. 

The results from the number of therapeutic sessions are interesting, but the conclusions seem a bit off. We must consider that 8-14 sessions are the usual number of sessions in Exposure Therapy, and therefore we shouldn't assume that more sessions would lead to better outcomes. Treating PTSD is challenging and it continues to be a struggle for clinicians to find a good way to address this clinically.

As mentioned, only war PTSD patients were included, so conclusions should only be related to War PTSD and not others. PTSD is not exactly the same if we consider the traumatic event that led to its inception.  So perhaps the authors should consider focusing the review only on war PTSD and not extrapolating to the broader PTSD context.

Author Response

Thank you for your comments. We have tried to revise our manuscript in accordance with your comments as much as possible.

Round 2

Reviewer 2 Report (New Reviewer)

The reviewer acknowledges the efforts of the authors to improve the quality of the manuscript and the methodological soundness. The manuscript could however benefit from more improvement in the methodology and in the discussion sections, but the reviewer considers that at this stage, what was done is enough to endorse the publication. Good luck on your research efforts. 

This manuscript is a resubmission of an earlier submission. The following is a list of the peer review reports and author responses from that submission.

Round 1

Reviewer 1 Report

Hi

this is very similar to a previous meta-analysis by Kothgassner.  I think your justification for repeating the meta-analysis isn't clear.  Similarly, you have diverged from your protocol without explanation.

Specific feedback follows:

This paper could do with English language editing to improve conciseness and intent of meaning.  Some initial examples from the beginning of the paper (I haven't listed them all).  Line 24...extra word, Lines 9, 28, 32...To date, it is worth noting, Since up to... are all unnecessary.

Lines 65-7...This sentence is hard to follow.  I didn't really understand your justification for completing the meta-analysis given the recent one your reference.

Line 72...You report an interest in comparing with active controls.  Your protocol states the same.  Why step outside this to report studies with inactive controls? (Line 94)

Line 81...Why limit the early date to 2010? 

Introduction should include a description of VR exposure therapy...is this with therapist guidance?  Or not?

Line 98...what does no restrictions in diagnosis mean?  Shouldn't they have PTSD?

Line 101: should this be depressive symptoms measured by BDI as per protocol?

Results: please report total n for included studies.

Figure 3: is the final inactive study missing the study name?

Discussion: Line 203...doesn't match the protocol which was to compare to active control.

Line: 208/9...why is this systematic review better?  I briefly reviewed the Kothgassner review and yours appears very similar.  What is the purpose of repeating the SR?

Line 216...You state 5/8 published in the last decade but your search limit didn't allow you to find anything published before 2010

Line 251...most or all?

Author Response

Thank you for your constructive comments. We have revised our manuscript in full compliance with your comment. 

Reviewer 2 Report

The aim of this paper was to perform a systematic review and a meta-analysis investigating the effects of VR-based exposure therapy for PTSD
symptoms as compared to active (e.g., conventional exposure therapy) or
inactive controls (e.g., wait-list).

This manuscript is well written and easy to read, and the topic is interesting, since Virtual reality (VR) technology could offer new opportunities for the development of innovative clinical research, assessment, and intervention tools. Moreover, virtual reality is a potential digital solution with growing capacity to present customizable visual, auditory, tactile, or olfactory stimuli and demonstrate positive and important changes in treatments for trauma and stress disorders (problems with prevalence to increase), allowing a safer control of the conditions and a more personalized and customized follow-up by the therapists.

However, in their abstract, the authors said “little is known about the efficacy of VR-based exposure therapy for PTSD, but there are already several reviews and meta-analysis that address this topic (see below). What does this study bring new?

-       Deng, W., Hu, D., Xu, S., Liu, X., Zhao, J., Chen, Q., Liu, J., Zhang, Z., Jiang, W., Ma, L., Hong, X., Cheng, S., Liu, B., & Li, X. (2019). The efficacy of virtual reality exposure therapy for PTSD symptoms: A systematic review and meta-analysis. Journal of affective disorders, 257, 698–709. https://doi.org/10.1016/j.jad.2019.07.086.

-       Eshuis, L. V., van Gelderen, M. J., van Zuiden, M., Nijdam, M. J., Vermetten, E., Olff, M., & Bakker, A. (2021). Efficacy of immersive PTSD treatments: A systematic review of virtual and augmented reality exposure therapy and a meta-analysis of virtual reality exposure therapy. Journal of psychiatric research, 143, 516–527. https://doi.org/10.1016/j.jpsychires.2020.11.030.

-       Vianez, A., Marques, A., & Simões de Almeida, R. (2022). Virtual Reality Exposure Therapy for Armed Forces Veterans with Post-Traumatic Stress Disorder: A Systematic Review and Focus Group. International journal of environmental research and public health, 19(1), 464. https://doi.org/10.3390/ijerph19010464.

-       Kothgassner, O. D., Goreis, A., Kafka, J. X., Van Eickels, R. L., Plener, P. L., & Felnhofer, A. (2019). Virtual reality exposure therapy for posttraumatic stress disorder (PTSD): a meta-analysis. European journal of psychotraumatology, 10(1), 1654782. https://doi.org/10.1080/20008198.2019.1654782.

I consider “Introduction” section interesting, and the references are sufficient and up to date.

In Methods, line 77, authors wrote PRISAM instead of PRISMA. In “Eligibility Criteria”, why choose DSM-IV, when there is already a 5th edition? Nevertheless, the authors followed the methodological guidelines. Perhaps the description could be more detailed (e.g. what search equation was used in each database?)

Regarding discussion, this article being published in a special theme (Virtual Reality-Based Cognitive Training for Cognitive Function and Psychological Symptoms) expected the authors to analyze possible impacts on cognitive impairments. Also, what are the clinical and social implications of this paper? I recommend that the clinical and social implications of this study emerge in a more pragmatic way as well as what this study brings to the existing literature.

Author Response

(The authors gave the same response as above.)

Round 2

Reviewer 1 Report

I have reservations about completing further meta-analyses in this area...two were completed in 2019 and a further repeated late in 2021.  Although there are subtle differences between these reports, I'm not sure this analysis brings new knowledge to the area.